# Controversies and Perspectives of Time-Qualified Dietary Interventions

**DOI:** 10.3390/nu17243894

**Published:** 2025-12-12

**Authors:** Sofia Lotti, Silvia Gallosti, Ramona De Amicis, Simona Bertoli, Barbara Colombini, Gianluigi Mazzoccoli, Monica Dinu

**Affiliations:** 1Department of Experimental and Clinical Medicine, University of Florence, 50134 Florence, Italy; sofia.lotti@unifi.it (S.L.); barbara.colombini@unifi.it (B.C.); monica.dinu@unifi.it (M.D.); 2International Center for the Assessment of Nutritional Status and the Development of Dietary Intervention Strategies (ICANS-DIS), Department of Food, Environmental and Nutritional Sciences (DeFENS), University of Milan, 20133 Milan, Italy; silvia.gallosti@unimi.it (S.G.); simona.bertoli@unimi.it (S.B.); 3IRCSS Istituto Auxologico Italiano, Obesity Unit and Laboratory of Nutrition and Obesity Research, Department of Endocrine and Metabolic Diseases, 20145 Milan, Italy; 4Chronobiology Laboratory, Fondazione IRCCS Casa Sollievo della Sofferenza, Cappuccini Avenue, 71013 San Giovanni Rotondo, Italy; g.mazzoccoli@operapadrepio.it

**Keywords:** circadian rhythm, chrononutrition, time-restricted feeding, intermittent fasting, fasting-mimicking diet, metabolic health

## Abstract

Time-qualified dietary interventions, including time-restricted eating (TRE), intermittent fasting (IF), and periodic fasting-mimicking diets (FMDs), have emerged as strategies to improve metabolic health. While preclinical studies consistently demonstrate robust effects on energy metabolism, cardiometabolic function, and longevity, translation to humans remains heterogeneous. In free-living settings, most metabolic improvements observed with TRE and IF appear primarily driven by spontaneous caloric restriction rather than meal timing per se, and isocaloric randomized controlled trials generally show no additional benefits compared to standard calorie restriction. Evidence supporting circadian-specific advantages, particularly for early TRE, is promising but inconsistent and often context-dependent. Important uncertainties also persist regarding long-term efficacy, lean mass preservation, safety in specific populations, and the physiological impact of extended fasting windows. Despite these controversies, time-qualified diets represent a paradigm shift in nutritional science by integrating chronobiology with dietary patterns. Future directions include tailoring eating windows to individual chronotypes, combining fasting regimens with high-quality dietary patterns and structured physical activity, and clarifying the molecular mechanisms that may mediate calorie-independent benefits. Large, long-term, mechanistically informed human trials are essential to determine whether aligning eating behaviors with circadian biology can produce durable clinical improvements. Such work will ultimately shape the role of personalized chrononutrition in preventive and therapeutic nutrition.

## 1. Introduction

Across biological systems, from unicellular organisms to mammals, metabolic functions display robust 24 h oscillations that reflect the activity of the circadian timing system [1,2]. This endogenous clock has evolved to anticipate predictable environmental cycles, optimizing physiological processes such as sleep–wake behavior, hormone secretion, and nutrient metabolism. However, modern lifestyles characterized by prolonged light exposure, shift work, irregular sleep, and continuous food availability promote circadian misalignment, or “chronodisruption”, a condition associated with impaired metabolic homeostasis and increased cardiometabolic risk [3,4].

Among the environmental cues that synchronize circadian rhythms, feeding–fasting cycles play a pivotal role, particularly for peripheral clocks in metabolic tissues. This recognition has stimulated growing interest in dietary approaches that manipulate the timing of eating rather than its caloric content or macronutrient composition. Collectively termed time-qualified dietary interventions, these include time-restricted eating (TRE), various intermittent fasting (IF) regimens, and periodic fasting-mimicking diets (FMDs) [5,6,7]. Advances in chrononutrition have shown that human metabolic processes respond dynamically to nutrient timing, reinforcing the hypothesis that synchronizing food intake with circadian rhythms may confer health benefits beyond caloric restriction alone [8,9,10]. Yet, despite compelling mechanistic evidence and striking benefits observed in animal models, including protection against obesity, cardiometabolic dysfunction, and hepatosteatosis [11,12,13], human findings remain mixed. Some clinical studies report improvements in insulin sensitivity, glycemic control, and blood pressure independent of weight loss, particularly with early TRE [14,15], whereas others show minimal or no differences compared with standard calorie-restricted diets when energy intake is controlled [16,17,18]. These discrepancies have intensified debate surrounding whether timing exerts a meaningful metabolic effect in humans or primarily serves as a behavioral tool to reduce caloric intake.

This review critically evaluates the evidence on time-qualified dietary strategies through the lens of circadian biology. After summarizing the organization of the circadian timing system and its influence on metabolic pathways, we examine clinical findings across major time-targeted dietary models, with particular attention to controversies regarding the relative contributions of caloric restriction, timing, circadian alignment, and diet quality. Finally, we outline key gaps, methodological challenges, and future directions for precision chrononutrition and translational research.

## 2. Circadian Timing System and Molecular Clockworks in Metabolic Regulation

Living organisms rely on a complex circadian system to anticipate and adapt to daily environmental cycles [19,20]. In mammals, this system coordinates behavioral and metabolic rhythms through a hierarchical organization that integrates central and peripheral timekeeping mechanisms. Understanding how these clocks interact with feeding–fasting cycles is essential to contextualize the rationale behind time-qualified dietary interventions.

### 2.1. The Circadian Timing System

Light is the dominant environmental cue (zeitgeber) for the central circadian pacemaker located in the suprachiasmatic nuclei (SCN) of the hypothalamus. Intrinsically photosensitive retinal ganglion cells containing melanopsin transmit photic information to the SCN through the retinohypothalamic tract, initiating entrainment via glutamatergic signaling [21,22]. In turn, the SCN orchestrates daily rhythms in physiology and behavior by regulating neural and hormonal outputs, including melatonin and glucocorticoids, and by modulating autonomic tone [23,24].

Although the SCN is predominantly synchronized by light, peripheral clocks in tissues such as liver, adipose tissue, muscle, and gut exhibit strong responsiveness to feeding–fasting cycles. Nutrient availability, insulin secretion, and metabolic cues can shift the phase of peripheral oscillators independently of the SCN [25,26]. When feeding occurs at biologically inappropriate times, such as during the usual rest phase, peripheral organs may desynchronize from the centrally entrained SCN, leading to metabolic disturbances collectively referred to as circadian misalignment [27]. This misalignment is increasingly recognized as a contributor to impaired glucose tolerance, insulin resistance, and cardiometabolic risk. Beyond this classical framework, the timing of nutrient intake can further modulate metabolic pathways, influencing insulin-mediated glucose uptake, lipid oxidation, and thermogenic responses. Considering these adaptive mechanisms is crucial to understand how feeding schedules interact with environmental cues and affect circadian–metabolic coupling.

### 2.2. The Circadian Molecular Clockwork

At the cellular level, circadian rhythms arise from an autoregulatory transcription-translation feedback loop (TTFL) [28]. The core positive limb consists of the transcription factors CLOCK and BMAL1, which heterodimerize and activate target genes containing E-box regulatory elements, including the negative regulators Period (Per1–3) and Cryptochrome (Cry1–2). PER and CRY proteins accumulate, dimerize, translocate into the nucleus, and inhibit CLOCK-BMAL1 activity, generating ~24 h oscillations in gene expression [29].

Auxiliary loops add robustness and precision. CLOCK-BMAL1 also induces the nuclear receptors REV-ERBα/β and RORα/γ, which rhythmically repress or activate Bmal1 transcription, thereby stabilizing the cycle [30,31,32]. Post-translational modifications of clock proteins, including phosphorylation, sumoylation, and ubiquitination, further shape circadian periodicity, allowing dynamic responses to metabolic signals such as the AKT-GSK3β pathway [33,34]. This molecular system drives the rhythmic expression of thousands of clock-controlled genes across peripheral tissues, many of which regulate metabolic processes including glucose homeostasis, lipid metabolism, mitochondrial function, and hormonal signaling (Figure 1) [35,36,37]. As a result, metabolic efficiency and endocrine responsiveness vary substantially across the day.

### 2.3. Circadian Control of Metabolic Pathways

Feeding–fasting cycles are among the strongest zeitgebers for peripheral metabolic clocks. Even in the absence of a functional SCN, timed feeding can entrain hepatic rhythms and synchronize transcriptional profiles of metabolic genes through nutrient-sensitive pathways such as CREB, SREBP, ATF6, and FoxO1 [38,39]. Conversely, inappropriate feeding timing, particularly during the evening or night, reduces the amplitude of metabolic oscillations and impairs glucose tolerance, lipid handling, and insulin sensitivity.

In humans, circadian variation in metabolic functions is well documented. Insulin sensitivity peaks in the morning and declines sharply by evening; identical meals lead to significantly higher postprandial glucose and reduced β-cell responsiveness at night [40,41]. This diurnal pattern provides a mechanistic framework for the hypothesis that early day eating may align better with endogenous metabolic rhythms, whereas late eating contributes to cardiometabolic impairment.

Comprehensive mechanistic analyses of circadian–metabolic interactions have been extensively reviewed elsewhere [42,43,44]. For the purposes of this review, the essential point is that the circadian system imposes temporal variation in metabolic efficiency. This biological architecture provides the conceptual basis for understanding why the timing of food intake can influence metabolic outcomes. While not all time-qualified dietary strategies are explicitly designed to achieve circadian alignment, their metabolic effects inevitably intersect with circadian regulation, which helps contextualize their potential benefits and limitations.

## 3. Time-Qualified Dietary Interventions

Time-qualified dietary interventions encompass a broad spectrum of nutritional strategies that manipulate the timing, duration, or periodicity of energy intake. Although their structures differ, they share the premise that when food is consumed, it may modulate metabolic outcomes alongside what and how much is eaten. They include practices ranging from religious fasting to daily TRE, to IF protocols that incorporate full or partial fasting days, as well as hybrid and emerging models. Table 1 summarizes their defining characteristics.

Despite their heterogeneity, these dietary patterns pursue related objectives: extending fasting duration to promote metabolic switching, ketogenesis, and improved substrate utilization, enhancing circadian alignment by consolidating food intake into periods of high metabolic efficiency, and simplifying dietary structure by limiting eating opportunities and thereby reducing caloric intake without explicit calorie counting. In the following sections, we focus on the most clinically relevant and well-studied models within this spectrum, emphasizing those supported by controlled human trials and emerging evidence in metabolic and circadian physiology.

## 4. Time-Restricted Eating (TRE)

TRE, also referred to as time-restricted feeding (TRF) in animal studies, is characterized by the confinement of daily caloric intake to a consistent eating window, generally ranging from 4 to 12 h, without explicit prescription of caloric reduction or dietary composition [45,46]. A typical protocol involves fasting for 16 h and consuming all meals within an 8 h period, although wider windows such as 14:10 or narrower ones such as 18:6 and 20:4 have also been explored [47]. Unlike traditional diets centered on energy restriction or macronutrient manipulation, TRE primarily targets the timing of food intake, aiming to realign eating behavior with circadian metabolic rhythms. In principle, this alignment favors the consolidation of food intake during the biologically active phase, generally daytime in humans, when insulin sensitivity, β-cell responsiveness, and metabolic efficiency peak [48]. In this context, early TRE (eTRE), which restricts intake to morning or early afternoon hours, has been postulated to offer physiological advantages over late TRE (lTRE), where food intake is shifted toward the evening.

### 4.1. Weight and Metabolic Effects

In animal models, restricting food intake to the active phase exerts remarkably robust effects on metabolic outcomes, protecting against diet-induced obesity, hepatic steatosis, and metabolic syndrome even in the absence of caloric restriction [49,50,51]. These findings have driven enthusiasm for the translational potential of TRE; however, human data have been considerably more heterogeneous.

Across randomized controlled trials and recent meta-analyses, TRE in adults with overweight or obesity produces modest but consistent reductions in body weight, fat mass, waist circumference, fasting insulin, and blood pressure, while effects on fasting glucose and HbA1c are smaller and often nonsignificant [52,53,54,55,56]. Importantly, these benefits appear to occur predominantly in ad libitum conditions, where TRE reliably induces a spontaneous reduction in daily caloric intake, typically in the range of 300 to 500 kcal per day [52]. This behavioral effect is now recognized as the principal driver of weight loss in most TRE studies.

The central question raised by these findings is whether TRE confers metabolic advantages independent of caloric restriction. Evidence from controlled feeding trials suggests that such timing-specific effects are limited. In isocaloric studies, where caloric intake is matched between TRE and control groups, differences in weight loss and cardiometabolic outcomes largely disappear [17]. Meta-analyses have confirmed that when compared with structured calorie-restricted diets, TRE does not produce substantially greater reductions in body weight, blood pressure, glucose, or lipid levels [18,56].

Nevertheless, some studies, particularly those employing eTRE under controlled laboratory conditions, report improvements in insulin sensitivity, β-cell responsiveness, and 24 h glucose dynamics independent of weight loss [14,15,57]. These results suggest that timing may exert modest but physiologically meaningful effects under specific circumstances. Yet, such improvements have not consistently translated into clinically relevant advantages in free-living individuals, particularly when diet quality and caloric intake are optimized. The inconsistency across trials remains a major point of controversy and highlights the need for mechanistic and long-term human studies capable of disentangling the relative contributions of timing, energy balance, and diet quality.

### 4.2. Early vs. Late Eating Windows

Several intervention studies have demonstrated that concentrating caloric intake earlier in the day, when metabolic efficiency is higher, can improve fasting insulin, glycemic excursions, and blood pressure more effectively than eating later [15,58]. These effects are supported by mechanistic evidence showing that evening meals occur at a time of reduced insulin sensitivity, diminished β-cell responsiveness, and elevated melatonin levels, all of which impair glucose tolerance [59,60,61]. Meta-analyses further indicate that eTRE may provide greater improvements in HOMA-IR and fasting glucose than lTRE, even when total caloric intake is not different between groups [62,63].

However, recent high-quality trials challenge the magnitude and generalizability of these timing effects. A large randomized controlled trial comparing early, late, and self-selected eight-hour eating windows, all combined with Mediterranean diet (MD) counseling, detected no differences in visceral adipose tissue reduction between TRE schedules or between TRE and usual care [64]. Likewise, in a highly controlled crossover study in women with overweight or obesity, both eTRE (08:00–16:00) and lTRE (13:00–21:00) produced no improvements in insulin sensitivity, glycemia, lipid profile, oxidative stress, or inflammatory markers under isocaloric conditions, despite excellent adherence [65]. In parallel, a recent 10 h TRE intervention in individuals at high risk for type 2 diabetes yielded no clinically meaningful weight loss or glycemic improvement, and benefits were not sustained over time [66].

Taken together, these findings suggest that while meal timing may modulate metabolic physiology under controlled laboratory conditions, its effects in free-living populations are substantially attenuated when caloric intake is matched, dietary quality is high, or timing interventions must compete with social and behavioral constraints. The translational relevance of early–late distinctions remains an open question, and it is likely that interindividual factors such as chronotype, habitual eating patterns, and socioeconomic context play a larger role than initially appreciated.

### 4.3. Adherence and Side Effects

The long-term feasibility of TRE is a recurrent concern. Short-term trials frequently report high adherence rates, yet observational and behavioral studies indicate that sustained compliance may decline when fasting periods interfere with work schedules, social commitments, or habitual eating patterns [67,68]. eTRE, despite potential metabolic advantages, is often perceived as socially restrictive, whereas lTRE may be more compatible with daily routines but less aligned with circadian physiology. Chronotype appears to influence adherence, with morning types finding eTRE more tolerable and evening types struggling with early food cutoffs, thus underscoring the importance of personalized approaches [9].

From a safety perspective, TRE is generally well tolerated. Reported adverse effects, when present, are usually mild and transient, including hunger, lightheadedness, and irritability during the initial adaptation period. However, specific populations such as individuals with diabetes using hypoglycemic medications or pregnant women require medical supervision before initiating TRE due to the risk of hypoglycemia or insufficient energy intake [46]. Additional debate has emerged following an observational study associating self-reported <8 h eating windows with higher long-term cardiovascular mortality [55], though this finding is difficult to interpret due to reliance on a single 24 h dietary recall, potential reverse causality, and confounding by health status. These limitations caution against drawing causal inferences and emphasize the scarcity of long-term outcome data for TRE interventions.

Overall, TRE represents a promising but not universally effective strategy. It offers a simplified dietary structure and modest cardiometabolic benefits, particularly in settings of habitual overconsumption. Yet its long-term sustainability, magnitude of calorie-independent effects, and applicability across diverse populations remain areas of active investigation.

## 5. Intermittent Fasting Regimens (IF)

IF encompasses a broad family of dietary approaches in which periods of substantial caloric restriction or complete fasting alternate with days or periods of unrestricted eating. Unlike TRE, which reorganizes caloric intake within a single day, IF protocols extend the fasting window to 24 h or longer, thereby imposing more pronounced fluctuations in energy availability. The two most extensively studied models are alternate-day fasting (ADF) and the 5:2 diet, although variations such as periodic 24 h fasts and modified fasting approaches also exist [69,70]. These regimens are hypothesized to induce “metabolic switching”, a shift from glucose to lipid and ketone utilization, while simultaneously generating a caloric deficit that fosters weight loss. Their growing popularity has been matched by increased scientific interest, particularly regarding whether IF provides advantages beyond those predicted by energy restriction alone.

### 5.1. Efficacy for Weight Loss

Across randomized controlled trials, IF consistently induces modest but clinically relevant weight loss in individuals with overweight or obesity. Notably, its effectiveness is generally comparable to that of continuous caloric restriction (CER). Numerous recent meta-analyses, including large network analyses, have concluded that IF, CER, and TRE achieve similar reductions in body weight, with ADF occasionally demonstrating slightly greater weight loss yet only marginally so in absolute terms [56,71]. For example, in one network meta-analysis including 99 randomized trials, ADF produced approximately 1.3 kg more weight loss than CER, whereas both the 5:2 diet and TRE were statistically indistinguishable from CER in direct comparisons [56].

Longer-term trials, extending beyond six months, reinforce this conclusion: IF generally results in weight reductions of 2–6%, similar to those observed with CER and with no consistent evidence of superiority for any specific IF protocol [72]. In addition, weight loss tends to plateau after 8–12 weeks, suggesting that extending fasting durations or intensifying fasting days does not necessarily yield greater metabolic benefits [18]. Importantly, the slightly larger weight loss occasionally seen with ADF likely reflects the substantial caloric deficit created on fasting days rather than any inherent metabolic advantage of fasting itself. When energy intake is matched between IF and CER, differences in weight or fat mass loss largely disappear. ADF and the 5:2 diet therefore appear to function primarily as alternative behavioral structures that facilitate energy restriction, rather than as physiologically superior strategies.

### 5.2. Metabolic Health Effects

IF has been proposed to improve a broad array of metabolic markers, including insulin sensitivity, glucose homeostasis, lipid profiles, and blood pressure. However, human evidence suggests that these improvements are generally modest and largely proportional to weight loss, with little support for unique metabolic effects independent of energy restriction. Recent meta-analyses consistently show that while ADF may modestly reduce triglycerides, non-HDL cholesterol, and waist circumference, the effects of other IF formats such as the 5:2 diet or periodic 24 h fasting on glycemia or lipid profiles are small and often not statistically significant [56,71].

Similarly, umbrella reviews conclude that IF yields mild reductions in fasting insulin, HOMA-IR, and diastolic blood pressure, but has limited effects on fasting glucose, LDL cholesterol, or HDL cholesterol, and overall evidence quality is moderate to low [73]. Mechanistic explanations proposed for fasting-induced benefits, such as enhanced autophagy, mitochondrial remodeling, or neurotrophic signaling, are strongly supported in animal models but remain insufficiently demonstrated in humans [69,74]. Recent meta-analyses have therefore reiterated that the observed cardiometabolic improvements are best explained by energy deficit rather than fasting-specific metabolic pathways [18]. Thus, although IF clearly provides a viable alternative strategy for weight and metabolic risk reduction, current evidence does not indicate that it offers advantages over traditional calorie restriction when energy intake is controlled.

### 5.3. Challenges, Tolerability, and Suitability

As with TRE, adherence is one of the principal limitations of IF. Fasting days are often accompanied by hunger, irritability, headaches, fatigue, and reduced concentration, all of which contribute to higher dropout rates, particularly in ADF, where fasting periods extend to 24 h or longer [67,68]. The 5:2 diet generally shows better adherence and does not necessarily lead to compensatory overeating on non-fasting days [46,72], yet it still requires substantial behavioral adjustment, and its long-term sustainability remains uncertain. Moreover, the overall quality of evidence supporting IF is constrained by short intervention durations, small sample sizes, and considerable methodological heterogeneity across studies [73,75].

In addition, IF is not appropriate for all individuals. It is contraindicated during pregnancy, childhood and adolescence, in those with active eating disorders, and in certain chronic illnesses. Patients with diabetes who use glucose-lowering medications should attempt IF only under clinical supervision due to the risk of hypoglycemia. Some clinical guidelines such as the Canadian Adult Obesity Clinical Practice Guidelines now include IF as one option for weight management, acknowledging its comparable efficacy to CER while also noting its higher risk of attrition and lack of metabolic superiority [76].

Overall, the evidence base supporting IF remains weaker than that for more established dietary patterns such as the MD, which consistently demonstrates superior outcomes in long-term cardiometabolic trials [75]. For some individuals, IF may offer a psychologically appealing structure that simplifies decision-making around food, whereas others find the stringency of fasting days unsustainable. As with TRE, personal preference, lifestyle compatibility, and long-term adherence potential are likely to determine its practical utility.

## 6. Fasting-Mimicking Diet (FMD)

FMD is a periodic, low-calorie, low-protein, high-unsaturated fat dietary protocol designed to reproduce key metabolic features of prolonged fasting while still providing essential micronutrients and a modest energy supply. According to foundational protocols, the regimen provides approximately 4600 kJ (1100 kcal) on day 1 (11% protein, 46% fat, 43% carbohydrate) and ~3000 kJ (717 kcal) on days 2–5 (9% protein, 44% fat, 47% carbohydrate) [5,77]. The diet typically consists of vegetable soups, nuts, olives, teas, and specific formulated bars, intentionally designed to be low in protein and sugars but relatively higher in unsaturated fats. By markedly reducing total energy intake and limiting specific amino acids over several consecutive days, the FMD aims to induce a physiological fasting state characterized by ketogenesis and reduced insulin/IGF-1 signaling while still providing minimal energy to maintain basic physiological functions.

### 6.1. Evidence from Human Studies

Early human trials suggest that FMD may improve several metabolic risk factors, although the evidence base remains limited and heterogeneous. In a pivotal randomized clinical trial involving 100 generally healthy participants, three monthly cycles of FMD reduced body weight by an average of 2.6 kg, primarily through reductions in total and trunk fat, without significantly affecting lean body mass [5]. The intervention also lowered systolic blood pressure and IGF-1 levels. Post hoc analyses indicated that these benefits were most pronounced in individuals with elevated baseline IGF-1, glucose, or *C*-reactive protein, suggesting that FMD may exert targeted benefits in metabolically at-risk individuals rather than providing uniform advantages across healthy populations [5].

Additional evidence supports the potential of FMD to preserve lean mass relative to continuous dietary approaches. In a study comparing four cycles of FMD with four months of a continuous MD, both interventions reduced weight and cardiometabolic risk markers; however, meaningful differences emerged in body composition [78]. The MD group exhibited a significant loss of fat-free mass during follow-up, whereas the FMD group preserved lean mass. Notably, FMD cycles also produced sustained reductions in HbA1c and insulin that persisted for three months after the intervention, an effect not observed in the MD group, suggesting a possible “metabolic memory” associated with the periodic fasting pattern [78].

Similar findings were reported in women with obesity in a trial comparing FMD with CER [77]. Although weight loss was comparable between groups, the CER group experienced significant reductions in basal metabolic rate (BMR) and muscle mass. In contrast, the FMD group preserved both BMR and muscle mass and exhibited a greater improvement in insulin resistance. Hormonal responses also diverged: CER increased serum levels of appetite-stimulating hormones such as ghrelin and neuropeptide-Y, whereas FMD did not—suggesting that FMD may offer better long-term appetite regulation [77].

Clinical applications of FMD have expanded into oncology, supported by preclinical evidence that fasting can enhance healthy-cell resistance to chemotherapy-induced toxicity while increasing cancer-cell vulnerability through differential stress responses. Early stage trials in patients with HER2-negative breast cancer receiving neoadjuvant chemotherapy showed that administering FMD for three days before and on the day of treatment significantly reduced chemotherapy-related toxicity, including high-grade vomiting and neutropenia [79]. Patients in the FMD group also demonstrated improved pathological and radiological tumor responses, along with reductions in IGF-1 and inflammatory markers such as hs-CRP [79]. Nevertheless, these results remain preliminary, and larger controlled trials are required before FMD can be integrated into routine oncology practice [80].

### 6.2. Controversies and Practical Issues

Despite growing interest, several controversies surround the use of FMD. Although animal and cellular studies demonstrate effects on longevity, autophagy, tissue regeneration, and metabolic health, robust data in humans remain limited [68,74]. Most clinical trials include fewer than 100 participants, are short in duration, and rely on surrogate outcomes rather than hard clinical endpoints. Recent work highlights that the growing popularity of FMD contrasts sharply with the relatively small number and modest quality of human studies, particularly in populations with complex diseases such as cancer [68,80]. In addition, it is unclear whether observed benefits are specific to the FMD formulation (low protein, relatively higher fat, targeted micronutrient composition) or reflect more general effects of periodic caloric restriction. While some evidence suggests that FMD may better preserve metabolic rate and lean mass than continuous restriction [77,78], direct comparisons with other IF regimens are still lacking.

Another practical concern is adherence and accessibility. Adherence to fasting protocols can be difficult for patients with chronic conditions. In a study evaluating safety and feasibility in people with multiple sclerosis, adherence to calorie-restriction protocols was generally poor, even with clinical support [81]. While FMD-type interventions are safe, the drop-out rates in clinical trials suggest that long-term compliance requires significant willpower or behavioral support [78,81]. For instance, in the comparison with the MD, the FMD group had a higher dropout rate (15.9% vs. 5.0%), frequently attributed to adverse events such as weakness or dissatisfaction with the provided food items [78]. High attrition has also been documented by Wei and colleagues [5].

There is also the important consideration of cost and commercialization. The FMD protocol is often delivered through a proprietary meal kit (L-Nutra), which can be expensive and may limit accessibility. This raises questions about long-term sustainability: will individuals consistently undertake a 5-day FMD cycle every month for years? Moreover, although the FMD appears to preserve lean mass more effectively than continuous dieting in head-to-head trials [77,78], the absolute protein intake during the 5-day cycle is very low, necessitating careful re-feeding strategies to ensure adequate muscle protein synthesis between cycles.

From a safety standpoint, FMD is not appropriate for everyone. It is generally contraindicated in individuals who are underweight or have unstable medical conditions. During the FMD phase, close clinical monitoring is advised for those using glucose-lowering or antihypertensive medications, as dose adjustments may be required to prevent hypoglycemia or hypotension. Notably, supervised FMD interventions in trials involving patients with type 2 diabetes have been conducted without major safety issues, but always under medical oversight [82]. Some concerns have also been raised regarding vascular responses: certain markers of endothelial function, such as the Reactive Hyperemia Index (RHI), have shown paradoxical decreases following FMD cycles [78]. Although the authors proposed that this pattern might reflect a “vascular rejuvenation” phenomenon similar to that observed in younger individuals rather than true dysfunction, this interpretation remains debated. These findings highlight the need for additional mechanistic research to determine whether periodic severe restriction has short-term effects on vascular reactivity, particularly in individuals with pre-existing cardiovascular disease.

In summary, the FMD represents a compelling intersection between nutrition and fasting biology. However, its role in clinical practice will ultimately depend on its comparative effectiveness relative to simpler dietary strategies, its long-term feasibility, and evidence from larger, rigorously designed trials.

## 7. Controversies, Gaps, and Future Perspectives

The rapid expansion of chrononutrition research has generated considerable enthusiasm for time-qualified dietary interventions as potential tools for improving metabolic health. However, the evidence accumulated over the past decade, particularly from controlled human studies, reveals important inconsistencies and unresolved questions. Although TRE, IF, and FMD have each demonstrated some degree of clinical potential, substantial uncertainty persists regarding the magnitude, mechanisms, and durability of their effects (Figure 2).

### 7.1. Timing Versus Energy Restriction

A central question is whether the metabolic benefits attributed to TRE stem from meal timing itself or from the reduction in energy intake that often accompanies it. Meta-analyses consistently show that, under free-living conditions, the modest but significant weight loss observed with ad libitum TRE is primarily driven by a spontaneous reduction in daily caloric intake rather than by meal timing per se [18,52,56]. When individuals eat freely within a restricted window (e.g., 8 h), they typically consume fewer calories than those with unrestricted eating schedules, even without explicit instructions to reduce intake. In this sense, TRE appears to function largely as a behavioral constraint on energy intake rather than as a metabolic intervention that enhances fat oxidation independently of caloric balance.

However, this spontaneous energy deficit is not necessarily effortless or physiologically neutral. Emerging evidence challenges the assumption that prolonged fasting naturally suppresses appetite. A recent systematic review and meta-analysis of randomized controlled trials found that TRE increases subjective hunger relative to isocaloric control diets [83]. These findings suggest that extended fasting intervals may activate compensatory orexigenic pathways, potentially involving ghrelin or other appetite-related hormones that could hinder long-term adherence or promote compensatory overeating once the eating window opens.

### 7.2. Evidence from Isocaloric Human Studies

Isocaloric studies provide the most direct evidence of whether meal timing independently influences metabolic outcomes. Across this body of work, data consistently show that when energy intake is matched, TRE does not confer additional weight-loss or metabolic benefits. The strongest evidence comes from the landmark 12-month randomized trial by Liu et al. [17], in which 139 adults with obesity were assigned to either CR alone or CR combined with an 8 h eating window (08:00–16:00). The study found no significant differences in weight loss (−8.0 kg vs. −6.3 kg), body fat, or metabolic risk factors between groups, demonstrating that time restriction offers no added benefit beyond daily caloric reduction [17].

Recent systematic reviews reinforce this conclusion. An analysis of energy-matched randomized controlled trials reported that adding TRE to a CR regimen did not produce greater weight loss or cardiometabolic improvements in most studies [52]. Similarly, a meta-analysis of eight randomized controlled trials involving 579 participants found that although TRE combined with CR reduced weight relative to a non-CR control, it yielded no significant improvements in blood pressure, glucose profile, or lipid profile compared with CR alone [18]. Mechanistic data have also challenged the hypothesis that TRE enhances nutrient absorption efficiency. A randomized crossover trial using bomb calorimetry to quantify stool energy loss found that eTRE did not alter intestinal energy or macronutrient absorption relative to a 12 h eating schedule, confirming that weight-loss effects are not attributable to malabsorption [84].

Findings in specific populations are consistent with this pattern. In endurance-trained athletes, a 4-week isocaloric crossover trial showed that a 16:8 TRE protocol did not affect resting energy expenditure (REE) or biomarkers of cardiometabolic disease risk compared with a typical diet, indicating that TRE does not “boost metabolism” as often claimed, but rather limits opportunities to eat [85]. More concerningly, some evidence points to potential drawbacks: a 12-week feeding trial reported that although combining TRE with a healthy low-carbohydrate diet reduced body mass index, the TRE component was specifically associated with a greater loss of soft lean mass (muscle) compared with the non-TRE group [86].

### 7.3. Calorie-Independent Metabolic Effects: Signal or Noise?

While isocaloric conditions almost eliminate the additive effect of TRE on weight loss, debate persists regarding whether meal timing confers independent metabolic benefits. Early work, including the widely cited proof-of-concept trial by Sutton et al., reported that eTRE improved insulin sensitivity, blood pressure, and oxidative stress in men with prediabetes even in the absence of weight loss, suggesting that circadian alignment (i.e., eating earlier in the day) may elicit metabolic advantages independent of reductions in adiposity [14]. However, these effects appear inconsistent and highly dependent on participant characteristics. A systematic review of eTRE interventions found that although eTRE can improve glycemic outcomes, such benefits are observed predominantly in healthy, normal-weight individuals, whereas results in overweight or prediabetic populations are more variable and often contingent on accompanying weight loss [58]. Thus, while meal timing may exert specific circadian-mediated effects on glycemic control, current evidence indicates that for most clinically relevant outcomes, including weight loss and cardiovascular risk reduction, calorie restriction remains the primary determinant.

Recent workshop reports from major health institutions, including the National Heart, Lung, and Blood Institute (NHLBI) and the National Institutes of Health (NIH), have also emphasized these uncertainties. They conclude that, despite growing public enthusiasm for chrononutrition, “rigorous evidence” isolating the metabolic effects of meal timing from caloric restriction in humans remains insufficient, representing a key research gap that currently limits the development of evidence-based precision nutrition guidelines [87,88].

### 7.4. The High-Fat Paradox in TRF

Much of the foundational preclinical evidence showing the powerful protective effects of TRF was generated in rodents fed a high-fat diet (HFD) to induce obesity. In those models, TRF robustly prevented the adverse consequences of the HFD. However, translating these findings to humans has proven elusive. Recent comprehensive reviews have highlighted a lack of human trials specifically testing whether TRE can mitigate the damage of a Western-style high-fat diet [87,88]. Unlike in mice, where timing seems to override diet quality, human data suggest that the metabolic harms of a poor diet cannot be simply “timed away”. For instance, a randomized controlled trial in healthy non-obese adults found that while isocaloric TRE reduced body weight, it failed to provide additional benefits on glucose homeostasis or lipid profiles compared to a control diet, suggesting that without an improvement in diet quality or substantial weight loss, timing alone is insufficient to drive metabolic health [89].

### 7.5. Interactions Between Meal Timing and Diet Quality

Emerging evidence suggests that the greatest benefits are likely achieved when time-related strategies are combined with high-quality dietary patterns. Research combining TRE with specific macronutrient compositions indicates that the eating window must be nutritionally optimized to support satiety and metabolic flexibility [90,91]. Pairing an early eating window with a nutrient-dense, plant-rich diet may create a synergistic effect, but future clinical guidelines must emphasize that meal timing strategies are intended to complement, not replace, recommendations for healthy food choices. For example, a recent trial showed that combining a Healthy Low-Carbohydrate Diet (HLCD) with TRE resulted in significant changes to the gut microbiome, specifically decreasing fecal branched-chain amino acids and enriching probiotic species. This specific metabolic signature was not observed with TRE alone, suggesting that diet quality is a prerequisite for maximizing the microbiome-mediated benefits of fasting [86].

### 7.6. Lean Mass Loss: A Persistent Concern

In the clinical management of obesity, a primary concern with any weight-loss diet is the quality of weight loss. This has emerged as a significant safety signal in recent TRE trials. A feasibility study in adults with overweight found that while TRE induced weight loss, a concerning 65% of that weight loss was appendicular lean mass, a proportion significantly higher than what is typically observed with standard caloric restriction [92]. Similarly, a rigorous trial comparing 4 h vs. 6 h TRE windows found that while both regimens reduced body weight and insulin resistance, they also led to significant reductions in lean mass, raising questions about the long-term safety of such restrictive windows for musculoskeletal health [93].

Even in active populations, where exercise typically protects muscle tissue, TRE may have detrimental anabolic effects. A randomized trial in resistance-trained males demonstrated that while an 8 h TRE window reduced fat mass, it significantly lowered testosterone and IGF-1 levels compared to a normal diet, potentially creating a catabolic environment that could hinder long-term muscle maintenance and performance [94]. These findings are supported by other data in middle-aged adults, where TRE was effective for weight loss but required careful monitoring to prevent sarcopenia [95,96].

Therefore, the uncritical recommendation of TRE for weight loss may carry hidden risks. To mitigate the loss of metabolically active tissue, clinical protocols must integrate resistance exercise and ensure adequate protein distribution, factors often neglected in simple “eat within this window” advice.

### 7.7. Diet Quality, Eating Behaviors, and Real-World Implementation

A recurring critique of TRE and IF studies is that many do not tightly regulate diet composition. It remains plausible that a low-quality diet consumed within a short eating window may offer fewer, or even detrimental effects compared with a balanced diet distributed across the day. For example, if individuals use TRE as an opportunity to binge on energy-dense, nutrient-poor foods during the feeding window, the net impact on metabolic health may be unfavorable. Thus, the interaction between what is eaten and when it is eaten represents a critical and still underexplored research domain.

Recent work highlights the importance of integrating chrononutrition with established healthy dietary patterns, such as Mediterranean or plant-rich diets, to maximize metabolic improvements [8,97,98]. Early evidence supports this combined approach: adherence to a MD aligned with early TRE or periodic fasting appears to amplify benefits, including improved lipid profiles and anti-inflammatory effects, beyond those achieved by either strategy alone [99,100]. Moving forward, clinical guidelines will likely emphasize that meal-timing strategies should complement, not replace, nutritious food choices and overall dietary quality.

### 7.8. The Impact of Individual Chronotype

Chronotype, defined as an individual’s natural preference for sleep, activity, and feeding timing, reflects the functioning of internal circadian clocks and shapes hormonal rhythms, appetite, and metabolic responses [9,101]. Despite its relevance, it is still rarely considered in dietary guidelines. This omission may be particularly problematic for evening chronotypes, who typically exhibit later activity and eating patterns and are more prone to weight gain, metabolic dysregulation, and other adverse health outcomes [97,102,103]. Evening types also tend to report poorer lifestyle habits and dietary behaviors [104,105].

These differences suggest that a “one-size-fits-all” approach to meal timing may not be optimal. Chronotype may influence an individual’s ability to adhere to specific TRE schedules and could even predict weight-loss or body-composition outcomes [106,107]. Morning types, who naturally prefer earlier meals, may find eTRE more compatible, whereas evening types might tolerate later eating windows better. Emerging evidence highlights that chronotype significantly influences eating behaviors and metabolic risk. Specifically, evening types have been associated with a higher BMI and a greater prevalence of maladaptive eating behaviors, such as Binge Eating Disorder, compared to morning types. Furthermore, maintaining diet quality appears more challenging for E-types, as they demonstrate lower adherence to healthy dietary patterns like the Mediterranean Diet. Therefore, aligning meal timing interventions with an individual’s chronotype is crucial not only for adherence but also to counteract the specific behavioral and metabolic vulnerabilities associated with eveningness [108,109]. Standard dietary advice emphasizing earlier energy intake may therefore conflict with the biological and behavioral tendencies of evening chronotypes, reducing adherence and effectiveness [110,111].

In this context, emerging evidence supports the potential of chronotype-aligned dietary strategies. In a randomized controlled trial, a hypocaloric diet matched to the participants’ chronotype produced larger reductions in body weight and fat mass than a non-adapted isocaloric diet [112]. Another clinical trial is currently testing whether a chronotype-adapted diet improves weight loss, cardiometabolic risk markers, and gut microbiome profiles more effectively than a conventional calorie-distributed diet [113]. The gut microbiome itself displays circadian oscillations, and disruptions in microbial rhythmicity are linked to impaired glucose control, altered lipid metabolism, and low-grade inflammation [114]. Aligning meal timing with circadian biology may therefore help preserve microbiome rhythmicity and support healthier host–microbe interactions.

Although current evidence remains preliminary, it increasingly suggests that tailoring time-related dietary interventions to individual biological and behavioral traits could confer additional benefits. Reflecting this, a recent scientific statement from the American Heart Association explicitly recommends that “individual chronotype should be considered in guiding the timing of interventions or treatment” [9]. Personalizing eating windows to an individual’s chronotype, lifestyle, work schedule, and behavioral preferences may enhance long-term adherence and improve the effectiveness of meal-timing strategies [111]. Genetic variation, including differences in clock genes and metabolic pathways, may further contribute to inter-individual variability in response [8]. These insights point toward the emerging field of precision chrononutrition, in which meal timing is aligned with the internal biological clock and potentially optimized through complementary strategies such as light management or melatonin modulation [9,101].

### 7.9. Long-Term Sustainability and Safety

Most TRE and IF trials have lasted between 2 and 6 months, with only a few extending to 12 months. Consequently, data on multi-year adherence and long-term health effects remain limited [71,72,115]. It is unclear whether individuals can maintain weight loss or glycemic improvements over 2–3 years of intermittent fasting, or whether outcomes will plateau or regress, as commonly observed with other dietary interventions. Questions also remain regarding the potential long-term adverse effects of chronic IF. Concerns have been raised about possible hormonal alterations [116], such as whether prolonged fasting might chronically elevate cortisol levels or affect reproductive hormone regulation. Although short-term studies in men have generally not shown substantial reductions in testosterone, and 8-week protocols in women have not reported severe menstrual disturbances, longer follow-up is needed to determine sustained effects [117]. Moreover, recent observational mortality data, while not establishing causality, underscore the importance of ongoing monitoring of cardiovascular health among individuals practicing fasting regimens [118]. Long-term sustainability also depends on the interplay between diet and overall lifestyle rhythms. Disruptions in daily routines and sleep quality—as observed during the COVID-19 lockdown—have been directly linked to weight gain and BMI increase, underscoring that dietary timing interventions must be integrated with sleep hygiene and lifestyle management to be effective long-term [119].

An additional aspect that requires consideration when evaluating long-term safety is the nutritional adequacy of time-qualified dietary protocols. TRE and IF may inadvertently reduce overall dietary quality or micronutrient intake, particularly when eating windows are short or mealtime frequency is low [52,93]. Restricting opportunities for food intake can lead to insufficient consumption of protein, fiber, essential fatty acids, and key micronutrients such as calcium, vitamin D, iron, folate, and B vitamins, especially in individuals with higher nutritional requirements. Moreover, inadequate protein distribution across meals during prolonged fasting windows may blunt muscle protein synthesis, thereby contributing to gradual losses in lean mass over time [5,77]. To date, few studies have systematically evaluated micronutrient status, diet quality indices, or protein adequacy during long-term adherence to TRE or IF, representing a major gap in the literature [90,91].

Long-term sustainability and safety must also be evaluated across diverse populations (Table 2). Particular caution is needed for older adults and patients in catabolic states (e.g., cancer, chronic inflammatory diseases). Prolonged fasting windows or protocols with very low energy availability may not provide sufficient per-meal protein to stimulate muscle protein synthesis, potentially exacerbating sarcopenia [92,93]. Among athletes, the key questions concern performance outcomes and preservation of lean mass. For patients with chronic conditions such as fatty liver disease or polycystic ovary syndrome (PCOS), it remains to be tested whether meal timing can meaningfully improve disease management. Encouragingly, IF is increasingly being studied within these varied contexts, but rigorous, population-specific research will be essential to ensure long-term safety and to guide clinical recommendations.

### 7.10. Mechanistic Uncertainties and Research Priorities

Despite significant advances in circadian biology, key mechanistic questions underlying time-based dietary interventions remain unresolved. Animal models clearly show that fasting triggers autophagy, ketone production, and shifts in gene expression that enhance cellular stress resistance [74]. However, determining whether similar processes occur in humans, and at what magnitude, remains a major research priority. Direct assessment of autophagy markers, stem cell activation, or other cellular renewal pathways could help validate several widely discussed claims regarding intermittent fasting and tissue regeneration.

Notably, recent mechanistic work in mice demonstrated that a high-fiber diet can mimic several molecular signatures typically associated with CR without reducing total caloric intake, suggesting that dietary composition may interact with fasting-related metabolic pathways in ways not previously recognized [120]. This highlights the need to consider both nutrient quality and meal timing in mechanistic frameworks. Growing interest also centers on how time-related diets influence the gut microbiome, which exhibits robust diurnal oscillations and plays a critical role in host metabolism [114,121]. Early studies suggest that TRE may help restore daily fluctuations in microbial populations and that prolonged fasting intervals may increase microbial diversity, changes that could confer downstream metabolic advantages [121,122,123]. Yet, this evidence remains preliminary. Further work is needed to elucidate how microbial rhythmicity interacts with hormonal signals such as ghrelin and leptin under different eating schedules, and whether these interactions meaningfully influence metabolic outcomes.

Collectively, these gaps underscore the need for mechanistic studies that integrate direct physiological and metabolic measurement rather than relying solely on anthropometric or biochemical endpoints. Future research should incorporate objective assessments of energy expenditure and metabolic responses, including resting energy expenditure, diet-induced thermogenesis, substrate oxidation via indirect calorimetry, and continuous physical activity monitoring. Because circadian misalignment can alter substrate utilization, sleep architecture, and thermogenic responses, studies should also integrate polysomnography or validated sleep trackers to evaluate sleep duration, fragmentation, and chronotype-specific timing. Additionally, neuroendocrine pathways regulating appetite and circadian feeding behavior should be systematically examined to clarify the central mechanisms mediating responses to timed eating.

### 7.11. Combination with Other Lifestyle Interventions

Another future direction involves integrating meal-timing strategies with other lifestyle interventions such as exercise and sleep optimization. The timing of exercise relative to meals—for example, fasted morning exercise versus fed evening exercise—may produce distinct metabolic adaptations. Preliminary studies suggest that exercising during the fasting period may enhance fat oxidation, but the most effective protocols and their long-term implications remain to be determined [124]. Likewise, ensuring sufficient and regular sleep in conjunction with timed eating may strengthen circadian alignment, potentially amplifying metabolic benefits [125]. More comprehensive lifestyle programs that incorporate light-exposure management, sleep hygiene, structured physical activity, and meal timing could provide a holistic approach to mitigating circadian disruption and reducing metabolic-syndrome risk.

## 8. Conclusions

Time-related diets are at a stage comparable to the early development of macro-nutrient-focused diets (e.g., low-fat, low-carbohydrate) decades ago: promising, but still working toward defining their optimal role. Their appeal stems from their conceptual simplicity, potential to improve metabolic health without explicit calorie counting, and alignment with fundamental principles of circadian physiology. Yet, despite substantial interest and an expanding body of research, the evidence supporting their efficacy and specificity in humans remains mixed and often contradictory.

Across controlled trials, modest improvements in weight, adiposity, fasting insulin, and blood pressure are consistently observed with TRE and several IF regimens. However, these benefits occur primarily in conditions of ad libitum intake and are strongly mediated by spontaneous reductions in daily energy consumption. When caloric intake is equalized, either through controlled feeding or structured dietary counseling, the advantages of meal timing largely disappear, highlighting the central role of energy balance in determining metabolic outcomes. Practically, these interventions can serve as tools to facilitate adherence to healthy eating patterns, reduce caloric intake, and structure meals, rather than as inherently superior metabolic therapies.

The FMD represents a conceptually distinct approach, seeking to reproduce the physiology of prolonged fasting through periodic, structured hypocaloric cycles. Preliminary studies suggest promising effects on IGF-1 reduction, inflammatory markers, and metabolic risk factors, yet data remain limited by small sample sizes and short durations. Key practical considerations include preserving lean mass, accommodating individual chronotypes, and ensuring adherence in real-life settings.

Several unresolved issues continue to shape the field. The extent to which circadian alignment exerts calorie-independent metabolic effects is uncertain. Lean mass preservation remains a concern in several fasting protocols. Adherence varies widely and is influenced by individual chronotype, social patterns, and lifestyle constraints. An additional limitation relates to the heterogeneity of the participants’ baseline dietary prescriptions driven by comorbidities such as diabetes, hypertension, or kidney disease. These therapeutic diets often impose specific macronutrient distributions, sodium restrictions, or medication adjustments, potentially confounding the isolated effects of eating-window manipulations. Moreover, most studies last fewer than 12–16 weeks, leaving the long-term sustainability, safety, and clinical relevance of these interventions largely unknown. Mechanistic pathways frequently invoked to explain fasting benefits such as autophagy, mitochondrial remodeling, and metabolic switching are well supported in preclinical models but remain poorly demonstrated in humans under realistic dietary conditions.

Despite these limitations, chrononutrition holds significant potential. Future applications should focus on personalized strategies integrating circadian phenotype, sleep–wake cycles, diet quality, and metabolic profile. Long-term, rigorously controlled trials are needed to clarify which populations may derive the greatest benefit, under what conditions, and through which mechanisms.

In summary, while time-qualified dietary interventions offer a promising and physiologically compelling framework, current evidence supports their use primarily as tools to facilitate energy reduction and behavioral structure rather than as inherently superior metabolic therapies. Their clinical value will depend on clarifying mechanisms, addressing adherence challenges, and targeting interventions to individuals most likely to benefit from circadian-aligned eating strategies.

## Figures and Tables

**Figure 1 nutrients-17-03894-f001:**
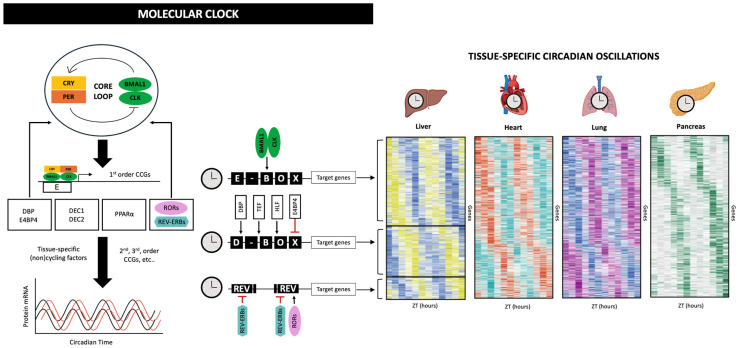
Schematic representation of the circadian molecular clockwork. The core CLOCK–BMAL1 transcriptional complex activates first-order clock-controlled genes, including Per and Cry, whose protein products provide negative feedback to inhibit CLOCK–BMAL1 activity. Auxiliary loops involving REV-ERBα/β and RORα/γ further stabilize rhythmic transcription. Together, these interconnected loops regulate the temporal expression of thousands of second-order clock-controlled genes that coordinate tissue-specific metabolic functions.

**Figure 2 nutrients-17-03894-f002:**
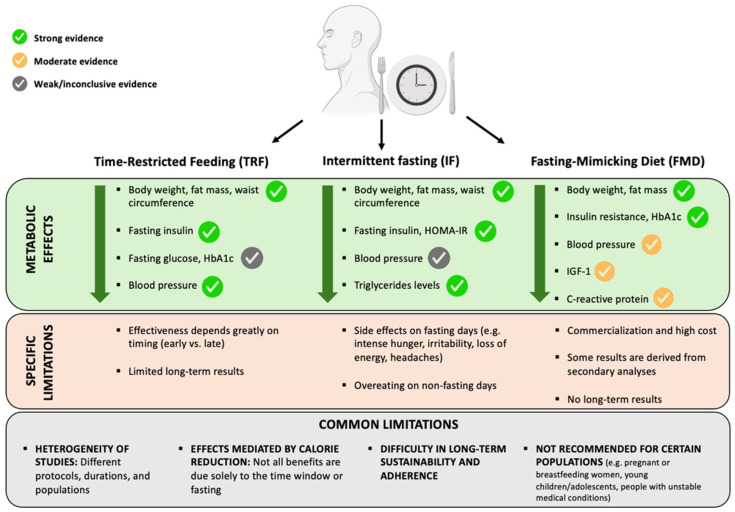
Comparison between time-restricted eating (TRE), intermittent fasting (IF) and fasting-mimicking diet (FMD) in terms of main metabolic effects and limitations. Colors indicate the strength of evidence (green indicates strong evidence, derived from reproducible findings across randomized controlled trials or high-quality meta-analyses with consistent outcomes; yellow reflects moderate evidence, characterized by positive but heterogeneous results, smaller samples, short intervention durations, or effects restricted to specific subgroups; grey denotes weak or inconclusive evidence, where results are contradictory, non-significant, or stem from exploratory or secondary analyses). Abbreviations: IGF-1 = insulin-like growth factor 1; HbA1c = glycated hemoglobin.

**Table 1 nutrients-17-03894-t001:** Major time-qualified dietary interventions described in the scientific literature.

Category	Interventions	Structure and Timing	Energy Pattern	Rationale
**Daily time-based restriction**	TRE/TRF	Eating window 4–12 h; fasting 12–20 h	Usually ad libitum	Align feeding with circadian rhythms; spontaneous energy reduction
	eTRE	Morning-early afternoon (e.g., 07:00–15:00)	Usually ad libitum	Enhances circadian alignment, improves insulin-glucose control
	lTRE	Afternoon-evening (e.g., 12:00–20:00)	Usually ad libitum	More practical socially; may misalign with circadian rhythms
	OMAD	Extreme TRE (1–2 h daily)	Ad libitum	Extreme daily fasting; long-term safety unclear
**Intermittent fasting (IF)**	ADF	Alternating fasting and feeding days	~0 kcal on fast days	Creates large intermittent energy deficits; metabolic switching
	mADF	Alternating days with ~500–600 kcal	~25% of energy needs on fast days	Better adherence while preserving fasting effects
	5:2 Diet	2 nonconsecutive low-calorie days/week	500–600 kcal on fasting days; unrestricted on 5 days	Weekly caloric deficit without daily restriction
	24 h periodic fasting	1–2 days per week	Water-only or very low calorie	Intermittent extended fasting without multi-day cycles
**Periodic multi-day fasting**	FMD	5-day cycles monthly or every few months	700–1100 kcal/day, low protein, high unsaturated fat	Mimics prolonged fasting (ketosis, IGF-1 reduction)
	Multi-day water fasting	2–7+ days	Minimal intake	Ketosis, autophagy; requires medical supervision
	Chemo-fasting	48–72 h pre- and peri-chemotherapy fasting	Water or very low kcal	Reduces toxicity by activating stress-resistance pathways (experimental)
**Hybrid and emerging approaches**	Chronotype-aligned eating	Eating window/energy distribution tailored to chronotype	Varies	Enhances adherence and circadian alignment
	TRE + IF combinations	TRE on non-fasting days	Mixed	Attempts to maximize fasting duration while improving practicality
	Macronutrient timing	Protein/carbohydrate timing (e.g., protein at breakfast)	Isocaloric	Targets circadian variation in macronutrient metabolism
**Religious fasting**	Ramadan	Dawn-to-sunset fasting; night eating	Usually ad libitum during night meals	Culturally structured fasting
	Greek Orthodox fasting cycles	Prolonged abstinence periods	Normal calories; restricted foods	Cultural fasting-abstinence pattern; reduced caloric density and modified macronutrients

**Abbreviations:** ADF, Alternate-Day Fasting; eTRE, Early Time-Restricted Eating; FMD, Fasting-Mimicking Diet; IF, Intermittent Fasting; lTRE, Late Time-Restricted Eating; mADF, Modified Alternate-Day Fasting; OMAD, One Meal A Day; TRE, Time-Restricted Eating; TRF, Time-Restricted Feeding.

**Table 2 nutrients-17-03894-t002:** Safety considerations and applicability of time-restricted eating strategies across different populations.

Population	Dietary Intervention	Positive Outcomes	Documented Risks/Concerns	Reference
**Adults with overweight/obesity**	FMD	↓ insulin resistance, ↔ BMR and lean mass	None reported	[77]
	TRE + Healthy Low-Carb Diet	↓ BMI	↓ soft lean mass	[86]
	TRE (ad libitum; ~8–9 h)	↓ body weight, ↓ number of eating occasions, ↓ lean mass and visceral fat	↓ ~3% lean mass lost	[92]
	TRE (4 h and 6 h)	↓ body weight, ↓ fasting insulin	↓ lean mass in both 4 h and 6 h TRE	[93]
**Athletes**	TRE 16:8 in endurance athletes	↓ total body fat mass, ↓ leg fat mass, ↔ fat-free mass	No improvement in REE and cardiometabolic markers	[85]
	TRE 16:8 in resistance-trained men	↓ fat mass, ↔ fat-free mass	↓ testosterone, ↓ IGF-1, ↔ REE and ↓ of respiratory ratio	[94]
**Prediabetes/High risk of Type 2 Diabetes**	eTRE (6 h)	↑ insulin sensitivity, ↓ blood pressure, ↓ oxidative stress, ↓ glycemic variability, ↓ evening hunger, ↔ body weight	Mild and transient side effects	[14]
	eTRE (9 h)	↑ post-meal glucose tolerance, ↓ fasting glucose, ↓ fasting triglycerides, ↔ insulin, gastrointestinal hormones, and free fatty acids; ↔ body weight	None reported	[57]
**Cancer**	FMD peri-chemotherapy cycles in HER2—breast cancer	↓ IGF-1, ↓ hs-CRP, ↓ toxicity (vomiting, neutropenia)	Requires clinical supervision; risk in undernourished patients	[79]
	TRE (<8 h)	No improvements	No significant association with cancer mortality, but ↑ risk of cardiovascular death	[118]

↓ indicates a decrease, ↑ indicates an increase, and the horizontal arrow indicates no significant change in the measured parameter. **Abbreviations:** BMI, Body Mass Index; BMR, Basal Metabolic Rate; eTRE, Early Time-Restricted Eating; FMD, Fasting-Mimicking Diet; hs-CRP, High-Sensitivity *C*-Reactive Protein; IGF-1, Insulin-like Growth Factor 1; REE, Resting Energy Expenditure; TRE, Time-Restricted Eating.

## Data Availability

No new data were created or analyzed in this study. Data sharing is not applicable to this article.

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
