# Peer review of "Controversies and Perspectives of Time-Qualified Dietary Interventions"

_nutrients, 2025, doi:10.3390/nu17243894_

Round 1

Reviewer 1 Report

Comments and Suggestions for Authors

The authors of the manuscript rightly describe the important topic of the impact of various dietary modifications related to time restriction eating on metabolism and health outcomes. The presented literature review is descriptive and narrative in nature, presenting the most important issues regarding both the methodology and the results obtained in currently available studies and meta-analyses in this field. The advantage is the critical approach and the indication of both the methodological shortcomings of these studies and the indication of further directions of scientific research.

However, for a better analysis of the issues described, it would be useful to summarize the characteristics and the most important results in the form of tables.

In addition, it would be worthwhile to expand on additional issues related to this research topic:

  • emphasizing the need to assess the nutritional value of dietary protocols and the risk of nutritional deficiencies, especially over a longer period of time,
  • emphasizing the dangers of energy and protein restrictions in individuals at risk of protein malnutrition, such as the elderly, or those in a state of catabolism resulting from, for example, cancer,
  • limitations related to other dietary recommendations resulting from illnesses among the study participants,
  • incorporating energy expenditure measurements into the research, including those related to thermogenesis and sleep quality, including respirometric tests, polysomnography, and assessment of hypothalamic function.

In the section devoted to explaining the potential mechanisms of time-qualified dietary interventions, more emphasis should be placed on factors related to the adaptation of energy metabolism to external factors, including nutrition.

The conclusions can be described in a more synthetic form, taking into account the practical recommendations resulting from the analysis.

Author Response

The authors of the manuscript rightly describe the important topic of the impact of various dietary modifications related to time restriction eating on metabolism and health outcomes. The presented literature review is descriptive and narrative in nature, presenting the most important issues regarding both the methodology and the results obtained in currently available studies and meta-analyses in this field. The advantage is the critical approach and the indication of both the methodological shortcomings of these studies and the indication of further directions of scientific research.

  • However, for a better analysis of the issues described, it would be useful to summarize the characteristics and the most important results in the form of tables.

We sincerely thank the reviewer for their valuable suggestion, which has helped us enhance the clarity of our manuscript. In response, we have added Table 2, summarizing the characteristics, safety considerations, and applicability of time-restricted eating strategies across different populations (lines 711-715). We chose not to include additional tables, as the metabolic outcomes and related limitations are comprehensively presented in Figure 2, and we aimed to avoid redundancy in the presentation of results.

  • In addition, it would be worthwhile to expand on additional issues related to this research topic:

  • emphasizing the need to assess the nutritional value of dietary protocols and the risk of nutritional deficiencies, especially over a longer period of time.

We thank the reviewer for this comment. We have added a section emphasizing the importance of evaluating the nutritional value of dietary protocols and the potential risk of nutritional deficiencies, particularly over extended periods (lines 687-697).

  • emphasizing the dangers of energy and protein restrictions in individuals at risk of protein malnutrition, such as the elderly, or those in a state of catabolism resulting from, for example, cancer.

We thank the reviewer for this comment. We have added the requested insight in the text and also the Table 2 (lines 700-703).

  • limitations related to other dietary recommendations resulting from illnesses among the study participants.

Thank you for the helpful comment. We have added this additional limitation (lines 789-793).

  • incorporating energy expenditure measurements into the research, including those related to thermogenesis and sleep quality, including respirometric tests, polysomnography, and assessment of hypothalamic function.

We thank the reviewer for the helpful comment. We have added the requested insight (lines 739-749).

  • In the section devoted to explaining the potential mechanisms of time-qualified dietary interventions, more emphasis should be placed on factors related to the adaptation of energy metabolism to external factors, including nutrition.

Thank you for the suggestion. We have added the requested insight (lines 99-103).

  • The conclusions can be described in a more synthetic form, taking into account the practical recommendations resulting from the analysis.

Thank you for the comment. We have shortened the conclusions and included more practical advice resulting from the analysis (lines 764-810).

Reviewer 2 Report

Comments and Suggestions for Authors

Thank you for the opportunity to review this manuscript. It is well-written and has valuable important insights into the topic.

Author Response

Thank you for the opportunity to review this manuscript. It is well-written and has valuable important insights into the topic.

 Thank you very much for the positive comment.

Reviewer 3 Report

Comments and Suggestions for Authors

Thank you for the article, I find it outstanding and utterly useful. The article will be valuable to researchers, clinicians, and nutrition professionals interested in chrononutrition and time-qualified dietary strategies. It is useful for my own knowlege !

Author Response

Thank you for the article, I find it outstanding and utterly useful. The article will be valuable to researchers, clinicians, and nutrition professionals interested in chrononutrition and time-qualified dietary strategies. It is useful for my own knowlege!

 Thank you very much for your generous feedback. We truly appreciate your thoughtful evaluation and we are glad to hear that the article may be useful for researchers and clinicians working in chrononutrition.